# The Open MatSci ML Toolkit: A Flexible Framework for Machine Learning in Materials Science

**Santiago Miret** *                                     *santiago.miret@intel.com*
*Intel Labs*

**Kin Long Kelvin Lee** *                                *kin.long.kelvin.lee@intel.com*
*Intel Accelerated Computing Systems & Graphics*

**Carmelo Gonzales**                                      *carmelo.gonzales@intel.com*
*Intel Labs*

**Marcel Nassar**                                         *marcel.nassar@intel.com*
*Intel Labs*

**Matthew Spellings**                                     *mspells@vectorinstitute.ai*
*Vector Institute*

**Reviewed on OpenReview:** *https://openreview.net/forum?id=QBMyDZsPMd*

## Abstract

We present the Open MatSci ML Toolkit: a flexible, self-contained, and scalable Python-based framework to apply deep learning models and methods on scientific data with a specific focus on materials science and the OpenCatalyst Dataset. Our toolkit provides: 1. A scalable machine learning workflow for materials science leveraging PyTorch Lightning, which enables seamless scaling across different computation capabilities (laptop, server, cluster) and hardware platforms (CPU, GPU, XPU). 2. Deep Graph Library (DGL) support for rapid graph neural network prototyping and development. By publishing and sharing this toolkit with the research community via open-source release, we hope to: 1. Lower the entry barrier for new machine learning researchers and practitioners that want to get started with the OpenCatalyst dataset, which presently comprises the largest computational materials science dataset. 2. Enable the scientific community to apply advanced machine learning tools to high-impact scientific challenges, such as modeling of materials behavior for clean energy applications. We demonstrate the capabilities of our framework by enabling three new equivariant neural network models for multiple OpenCatalyst tasks and arrive at promising results for compute scaling and model performance. The code of the framework and experiments presented in this is paper are publicly available at `https://github.com/IntelLabs/matsciml`.

## 1 Introduction

Recent years have seen great advances in applying advanced machine learning methods, especially novel deep learning methods, to scientific challenges that rely on computational modeling in the development of new physical systems (Krenn et al., 2022; Axelrod et al., 2022; Jumper et al., 2021). Modern deep neural networks trained to reproduce physical calculations, which are used to understand and optimize the behavior of real systems, can operate with high accuracy and are often orders of magnitude faster compared to methods based solely on human-derived calculations (Friederich et al., 2021; Chen & Ong, 2022; Fung et al., 2021).

---

* Equal contribution.

However, these representational capabilities usually come with significant engineering costs: The large variety of physical systems studied in chemistry, as well as the methods to represent those physical systems along with the data-driven paradigms used to predict their behavior, can make it cumbersome, time-consuming and error-prone to adapt models and tools to new datasets and applications. To mitigate these frequent challenges, we present the Open MatSci ML Toolkit, a flexible framework to develop deep learning techniques for chemistry and materials science applications.

Catalysts are essential components of chemical processes that help accelerate the rate of various chemical reactions. Catalytic materials design, especially the design of low-cost catalysts, remains an ongoing challenge that will continue to become more and more important for a variety of applications, including renewable energy and sustainable agriculture. The OpenCatalyst Project, jointly developed by Fundamental AI Research (FAIR) at Meta AI and Carnegie Mellon University's Department of Chemical Engineering, encompasses one of the first large-scale datasets to enable the application of machine learning (ML) techniques, with the full dataset containing over 1.3 million molecular relaxations of 82 adsorbates on 55 different catalytic surfaces (Chanussot* et al., 2021). The original release from 2019 has also been supplemented by subsequent updates in 2020 and 2022, along with an active leaderboard and annual competition (Tran et al., 2022).

The notable effort of providing high-quality data for catalytic materials is a major step forward in enabling ML researchers and practitioners to innovate on materials design challenges at great computational scales, and has already enabled the development of new geometric deep learning architectures ((Klicpera et al., 2020b) (Gasteiger et al., 2021)), some of which have been trained with nearly a billion parameters (Sriram et al., 2022).

While the toolkit of the original OpenCatalyst repository is very powerful, it incorporates a significant amount of complexity due to various interacting pieces of software: model definitions, functions for distributed training, and task abstractions. These components are often not self-contained, which makes it difficult for new ML researchers to navigate and interact with the repository, create new architectures or modeling methods, and run experiments on the dataset. To address these usability challenges, we introduce the Open MatSci ML Toolkit, a flexible and easy-to-scale framework for deep learning on the Open Catalyst Dataset. We designed the Open MatSci ML Toolkit with the following basic principles and compelling features:

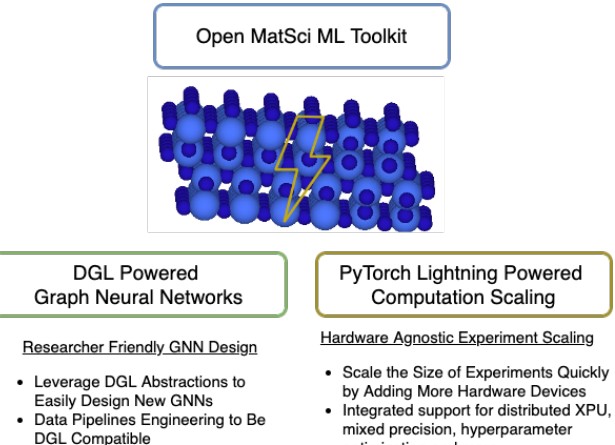

Figure 1: Open MatSci ML Toolkit enables researchers to create and perform scientific machine learning on the OpenCatalyst in a scalable manner leveraging DGL and PyTorch Lightning.

- Ease of use for new ML researchers and practitioners that want to get started with the OpenCatalyst dataset.

- Scalable computation leveraging PyTorch Lightning (Falcon & The PyTorch Lightning Team, 2019) across different computation capabilities (laptop, server, cluster) and hardware platforms (i.e. CPU, GPU, XPU) without sacrificing the scientific abstraction.

- Support for DGL (Wang et al., 2019) for rapid GNN development to complement the original repository's usage of PyTorch Geometric (Fey & Lenssen, 2019). In this work we showcase a set of models built on our toolkit for OCP-20 described in Section 4.

The examples outlined in this study and its associated open-source repository [1] show how to get started with the Open MatSci ML Toolkit using a simple Python script, Jupyter notebook, or the PyTorch Lightning CLI for a simple training instance on a subset of the original dataset (referred to as a development or dev-set) shipped with the repository that can be run on a laptop without needing to download and preprocess the minimal canonical dataset. Subsequently, we scale our example Python script to large compute systems with advanced equivariant deep learning models, including multi-GPU training on a single compute node, to distributed training across multiple nodes and GPUs in a computing cluster. Leveraging the capabilities of both PyTorch Lightning and DGL, we enable compute and experiment scaling with minimal additional overheard and complexity.

## 2 Background & Related Work

**Geometric Deep Learning (GDL)** generalizes neural networks to non-Euclidean domains such as graphs and manifolds (Bronstein et al., 2017; 2021). Such domains are increasingly used to model systems in scientific applications, such as molecular and crystal structures like those found in the OpenCatalyst project. In particular, GDL provides a way to represent entity interactions and various invariances and equivariances under geometric transformations—both vital for modeling molecules and catalyst interactions. Graphs and point cloud data[2] are the most common representations of such molecule-catalyst systems with different GDL methods operating on inputs with various inductive biases.

**Graph Neural Networks**, one of the earliest GDL applications to atomic systems, process a graph representation of molecular and solid-state material systems (Gilmer et al., 2017). These neural networks are able to take advantage of the natural representation of molecules as graphs, whereby message passing between nodes (atoms) resembles atomic interactions. Despite this, vanilla message passing GNNs are unable to distinguish between certain motifs and are bounded by the performance of the 1-Weisfeiler-Lehman (WL) isomorphism test (Xu et al., 2019). Even with recent advances in pushing GNNs beyond the WL-test (Morris et al., 2019), such networks do not utilize any of the known physical symmetries, thereby reducing their effectiveness on molecular data beyond their theoretical limitations.

**Equivariant Neural Networks** restrict the space of functions learnable by the network to obey symmetries found in the input data. Symmetries are a powerful inductive bias that drastically narrow the solution space that needs to be explored by the neural network, thereby making learning easier and improving generalization. Physical systems often exhibit many symmetries, either in their overall structure (e.g. water within the $C_{2v}$ point group), within structural motifs (e.g. three-fold rotation symmetry in methyl rotors), and, when nuclear dynamics are considered, complete nuclear permutation groups (Bunker & Jensen, 1998; Williams & Eisfeld, 2020) (e.g. ethane and the $G_{36}$ group (Mellor et al., 2019)). Composing many layers that individually respect symmetries relevant to a variety of problems has proven to be a powerful method to design deep neural network architectures for a range of modeling challenges. Exploiting such symmetries has emerged as a driving force for designing neural networks for physics models, for example as equivariance under the SO(3) group; that is, equivariance under translations and rotations in 3D space. Various works, such as Thomas et al. (2018); Anderson et al. (2019); Fuchs et al. (2020), leverage group representations to achieve full equivariance. While equivariant models can be more parameter efficient, they can also be computationally expensive or constrained in expressivity due to the model basis.

**Equivariant/Directional Graph Neural Networks** merge both graph and point cloud representations to leverage the benefits of both representations. One of these equivariant GNNs (E(n)-GNN by Satorras et al. (2021)) achieves this by separating node features into spatial components that are equivariant and invariant atomic features. These are then mixed together in a series of aggregation steps that preserve the spatial equivariance and feature invariance. Other GNNs take a more direct approach to equivariance by embedding angles, distance and dihedral angles in classic basis sets such as Legendre polynomials and spherical harmonics (GemNet (Gasteiger et al., 2021) and DimeNet (Klicpera et al., 2020a)), which possess well-known symmetry properties. These models can be viewed as directional GNNs since they typically expand their receptive field beyond the 1-hop neighborhood utilizing angular data to direct the message propagation. Directional

---

[1] https://github.com/IntelLabs/matsciml
[2] A collection of atomic particles in three-dimensional space, which unlike graphs do not have explicit connectivity.

models have high representational power, but come at a significant computational and memory cost. The performance of these models depends on the task and complexity of the data: EGNN can perform on-par with DimeNet and GemNet on simple classification tasks (Satorras et al., 2021); however, directional models typically perform better in more advanced applications such as conformer search (Ganea et al., 2021; Jing et al., 2022).

**Short-Range Equivariant Networks** restrict the ability of signals to travel long distances as layers are composed. In a typical graph neural network, signals from one node are able to travel an additional neighbor hop away in the graph for every message passing layer in the network; in contrast, short-range networks avoid transmitting these signals. These short-range restrictions can help promote localized representations that are more efficient to learn. One such architecture uses geometric algebra—or Clifford algebra—to formulate rotation-invariant and -equivariant combinations of input vectors, which are transmitted in a permutation-equivariant way *via* an attention mechanism (Spellings, 2021). The Allegro framework (Musaelian et al., 2022) learns to produce localized representations from a series of equivariant tensor products, rather than using geometric algebra to achieve rotation equivariance.

## 3 Software Framework

The Open MatSci ML Toolkit software framework is designed with great emphasis on abstraction and inheritance in order to maximize reusability and agility for machine learning researchers. These ideas are achieved in part by present-day best practices in Python as an object-oriented language, and through modern, specialized frameworks such as PyTorch Lightning and DGL. We believe these design choices make it significantly easier to apply novel model architectures and training techniques to scientific data, in particular the OpenCatalyst dataset. In addition to enabling researcher to construct graph neural networks with DGL as described in Section 3.2, we also enable support for point cloud representations, which are described in Section 3.3. These capabilities significantly the options for research to design diverse sets of model architectures that can leverage both graph and point cloud representations for atomistic data drawing upon some of the approaches we outline in Section 2. In the following sections, we will discuss changes in abstractions and refactoring steps from the original OpenCatalyst implementation.

### 3.1 PyTorch Lightning Refactor

In modern AI/ML workflows, the concept of "MLOps" comprises the lifecycle from model conception and implementation, training and testing in a variety of software/hardware environments, to drawing inferences on new data, and all of the iterative cycles in between. Thus, a non-negligible amount of time spent by researchers for new workloads is typically in engineering: interfacing data with new architectures, metric logging, performance profiling, and ensuring consistent functionality through the development process from testing locally on a laptop to distributed training on multiple computing nodes and across different types of accelerators. Because of the grand scale that OpenCatalyst aims for and successfully achieves, a large amount of the original codebase corresponds to performance and functionality; meaning that complexity is necessary to be able to take advantage of data parallelism, to perform hyperparameter optimization, and to support the various catalyst prediction tasks. This lays a significant amount of responsibility on both developers and users: the former must create a comprehensive suite of tests and rely heavily on CI/CD to ensure functionality, and the latter must navigate a maze of software dependencies and documentation, which are also maintained by the developer.

One half of the conceptual changes in the Open MatSci ML Toolkit—the other half being the primary graph framework—is to offload MLOps related components to a well-designed and maintained framework: PyTorch Lightning (Falcon & The PyTorch Lightning Team, 2019). By reusing dataset and framework components from OpenCatalyst and relying on the pipeline abstractions from PyTorch Lightning, we can maintain a more compact codebase while providing more flexibility, extendibility, transparency, and functionality. Figure 2 illustrates the end-to-end pipeline/directed acyclic graph for the Open MatSci ML Toolkit, whose elements should be relatively familiar to those who have used OpenCatalyst and/or PyTorch Lightning.

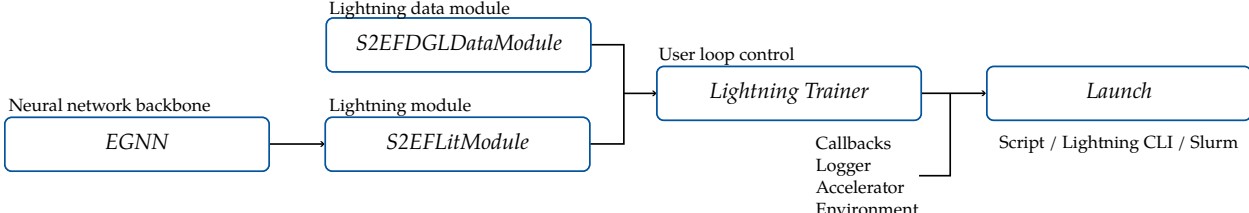

Figure 2: Illustration of the Open MatSci ML Toolkit pipeline with concrete components using the Structure to Energy & Forces ("S2EF") task. Dataset/task splits and configurations are specified through `LightningDataModules`. Task specific `LightningModules` encode the logic for training, metric logging, and how data is passed from dataset to an underlying abstract deep learning model. The `Trainer` interface provides an the ability to control feedback (e.g. logging, progress bars), training flow (`Callbacks`), and XPU usage without the need to modify the pipeline source code.

### 3.1.1 Data abstraction

In order to support future neural network research, we expanded the scope of the original OpenCatalyst dataset to support graph and non-graph data structures, as well as implemented a number of quality of life improvements to the general developer workflow. We refer the reader to the Appendix (i.e. Figure A.5) for more details pertaining to the changes, and here we only briefly highlight the core differences in user experience.

One of the core goals of the Open MatSci ML Toolkit is to have continuity from developing and testing on local environments—such as laptops—to using the pipeline in high performance computing environments. In terms of data pipeline abstraction, on the one end the Open MatSci ML Toolkit provides preprocessed, miniature (∼100 graphs) development or "devset"s: this circumvents the need to download, extract, and preprocess the data on personal computers constrained by storage and by computational power, while allowing researchers to prototype on the full pipeline. The development sets are created by taking random subsplits of the smaller data splits from the OpenCatalyst dataset. We also provide a mechanism for creating other splits not included with the Open MatSci ML Toolkit, facilitating further research into data efficiency. To use the devsets for development, there is a convenient mechanism for retrieving the DGL version of each task:

```
1 from ocpmodels.lightning.data_utils import S2EFDGLDataModule, IS2REDGLDataModule
2 # default settings optimized for local development; small batch, no parallel loaders
3 devset_module = S2EFDGLDataModule.from_devset()
```

On the other end of the spectrum, where one wishes to distribute the dataset across multiple workers on multiple compute nodes, users can leverage the same data modules as the miniature case: the `DistributedDataParallel` data sampling and loading is offloaded to PyTorch Lightning internals. We will discuss this further in Section 3.1.3.

As alluded to earlier in this section, the Open MatSci ML Toolkit data pipeline abstraction is designed to facilitate exploration of other data representations of materials systems, including algorithms less strongly linked to graph-based message passing. This includes recent approaches based on geometric algebra or Clifford algebras for point clouds by Spellings (2021), as well as methods based on tensor products, such as Musaelian et al. (2022). These algorithms operate locally on the bonds within a point cloud without typical message passing from one atom to another, while retaining the data efficiency imparted by model equivariance.

### 3.1.2 Model abstraction

The model abstraction, as seen in the bottom left nodes in Figure 2, comprises a neural network backbone and a task-specific `LightningModule`. In the concrete example described in Section 4, the `EGNN` model represents a subclass of an `AbstractEnergyModel`: a model that takes arbitrary input, and predicts the energy. For instance, a graph-based model will process nodes and perform some readout operation to reduce node and graph level features to a scalar value for the energy. At a higher level, the task specific `S2EFLitModule` is instantiated by passing an instance of `EGNN`, and implements the logic for training (i.e. forward-backward

passes), validation and testing, as well as logging. By conceptually separating the model (i.e. the neural network itself) from the training mechanism, researchers only need to focus on architecture development by subclassing `AbstractEnergyModel`, as the rest of the pipeline stays the same barring changes in *which kind of* data is required by the model.

### 3.1.3 Training loop

For typical regression tasks such as the initial structure to relaxed energy ("IS2RE"), the training loop takes advantage of automatic optimization in PyTorch Lightning, meaning that the backpropagation and optimization steps are abstracted away from the user. In cases with specialized autograd steps, such as the force component of "S2EF", and in many physics inspired neural network architectures, backpropagation is manually performed. While this highlights a gap in coverage by MLOps frameworks such as PyTorch Lightning, it also illustrates their open-endedness: one can maintain the flexibility necessary for research, while keeping the many other benefits and best practices guided by PyTorch Lightning.

With regards to changes in the behavior of the pipeline, users can rely on modular PyTorch Lightning components that can trigger at certain parts of the training loop—such as at the beginning of a training batch, or at the end of the validation epoch—with the `Callbacks` abstraction. The scope of such events include, but is not limited to: early stopping according to some validation metric, model checkpointing based on training intervals or metrics, as well as more specialized and flexible tasks like code profiling targeted to specific pipeline components—all without the need to inspect and modify the core source code.

Regarding logging/experiment tracking, the Open MatSci ML Toolkit relies on PyTorch Lightning `Loggers`, which are able to log locally to CSV or Tensorboard format, as well as to hosted services such as Weights and Biases and SigOpt; the latter of which can also be used for hyperparameter optimization (HPO), which makes up a valuable tool in a researcher's toolkit. Users are not confined to a single choice of logging or HPO platform, providing flexibility in tooling that fits a given researcher's workflow.

The final component relevant to the training process is the PyTorch Lightning `Trainer` class, which orchestrates the components mentioned above and executes training, validation, testing, and inference loops. The `Trainer` interface also configures more performance oriented settings such as accelerator usage, distributed compute, and mixed precision as shown in the example configuration below:

```
trainer = pl.Trainer(
    max_epochs=5,
    callbacks=[...],    # pipeline behavior;
    accelerator="gpu",  # use XPUs;
    precision="bf16",   # new data types;
    strategy="ddp",
    devices=8,          # 8 workers
    num_nodes=4         # per 4 nodes;
)
```

The main advantage is being able to seamlessly navigate between development and training cycles: the core pipeline remains unchanged, yet with a simple change in configuration, the user is able to take advantage of computational resources as they become available. Under the hood, the Lightning abstractions handle data movement to devices, autocasting in correct contexts, and orchestrating workers.

## 3.2 Deep Graph Library (DGL) Refactor

The original OpenCatalyst repo leverages Pytorch Geometric (PyG) for implementing various neural networks described in Section 2. In the Open MatSci ML Toolkit, we chose to complement the original implementation by building on top of the Deep Graph Library (DGL). While both PyG and DGL are highly performant libraries for graph neural networks, and the decision to choose one over the other is often subjective, we motivate our choice of DGL for this library as follows:

- *Graph Abstraction*: The DGL graph data structure `dgl.DGLGraph` offers more flexibility for storing diverse molecular data over the PyG structure `torch_geometric.data.Data`. This abstraction

allows for more general data pipelines amicable to experimentation and ablation studies; two qualities that are important in scientific exploration.

- *Cross-platform Optimization*: While both DGL and PyG are well-optimized for single-node GPU deployment, DGL also supports additional platforms, in particular efficient sparse matrix algorithms for CPUs with documented compute benefits on datacenter CPUs.

- *Support for Sampling-Based, and Distributed Training beyond Data-Parallel*: many applications involve large graph data that does not fit onto a single GPU. Such cases require specialized sampling techniques to either shrink the graph size or distribute the storage across multiple nodes. While both PyG and DGL support sampling-based training, DGL is more mature when it comes to sampling graphs and running distributed training of GNNs (Mostafa, 2021; Zheng et al., 2021).

Next, we discuss each of the above points in more detail.

### 3.2.1  Graph Abstraction

PyG's `Data` has a graph data structure composed of the following fixed attributes: node feature matrix `x`, edges `edge_index`, edge attributes `edge_attr`, labels `y`, and node position matrix `pos`. Additional attributes can be set using keyword argument collection in the constructor. In contrast, DGL's `DGLGraph` provides a dictionary-style access to graph data through `DGLGraph.ndata` (node features) and `DGLGraph.edata` (edge features). While, these dictionaries do not impose any restrictions on feature names or number, the `DGLGraph` object will enforce that the feature dimension match the number of nodes and edges.

While PyG's `Data` covers a large part of the use cases in the GNN world, we argue that using the flexible `DGLGraph` as example representation decouples the dataset from the model-specific data pipeline, while retaining data consistency such as matching the number of features and edges. This lends itself to a wide variety of customized pipelines that can explore various model explorations and ablation studies. For example, consider a molecular data structure that contains various features on both atoms and bonds: QM9 (Ramakrishnan et al., 2014), a common molecular property prediction benchmark, includes atom features detailing atom coordinates `x`, atom type `tp`, atomic number `z`, number of hydrogens `nH`, and hybridization Gilmer et al. (2017). A given set of molecular features would then contain the following fields:

```
class Example(NamedTuple):
  x: List[float]
  tp: List[bool]
  z: int
  nH: int
  bonds: List[Tuple[int, int]]
```

Assuming all the above fields have been cast into their appropriate `Tensor` types, converting `Example` to PyG's `Data` can be done as follows

```
example_dict = example._asdict()
src, dst = zip(*example_dict.pop('bonds'))
edge_index = torch.hstack((src, dst))
pos = example_dict.pop('x')

# assign to fixed attributes
embed_tp, embed_z, embed_nH = embed(example)
feat = torch.cat((embed_tp, embed_z, embed_nH)
data = Data(pos=pos, x=feat, edge_index=edge_index)
# assign to generic container
data = Data(pos=pos, edge_index=edge_index, **example_dict)
```

while in DGL's `DGLGraph` this would look like

```
src, dst = zip(*example_dict.pop('bonds'))
dgl_graph = dgl.graph((src, dst))
dgl_graph.ndata = example._asdict()
```

The above example demonstrates that the `DGLGraph` can accommodate diverse data without having to perform any special model-specific preprocessing while retaining graph consistency across assigned nodes and features. PyG `Data`, on the other hand, either requires feature preprocessing and assignment to the `x` field or generically attaching the features to the data object.

### 3.2.2 Cross-Platform Optimization

GPU acceleration plays a pivotal role in modern neural network training in general, including for GNNs. Recently, there has been an increasing interest in deploying GNNs, particularly in distributed computing settings, on other a diverse hardware platforms including datacenter CPUs and custom deep learning accelerators from nascent hardware vendors. While both DGL and PyG are well optimized for CUDA, DGL implements highly efficient kernels for sparse matrix operations central to graph processing through `libxsmm`, providing more flexibility in the type of hardware used for training and inference (Avancha et al., 2020).

### 3.2.3 Sampling and Distributed Training Beyond Data-Parallel

Increasingly, single device training of GNNs is reaching its limits as graphs and models scale in size and memory requirements. A common way to address these limits is to shrink the size of the graphs by sub-sampling the graph so that it fits into single device memory Hamilton et al. (2017). Yet, this type of sampling can lead to performance degradation due to the discarded neighborhood information, prompting a surge in new distributed training for GNNs (Hamilton et al., 2017; Md et al., 2021; Mostafa, 2021; Zheng et al., 2021). DGL already supports various forms of graph distributed training for both sampled and full batch training, and this capability could prove invaluable for molecular design datasets as both model and data complexity grows. While these types of distributed training might not be directly relevant to OpenCatalyst, they could be useful in future applications with large system sizes, such as large-scale molecular dynamics with billions of atoms (Musaelian et al., 2022; Guo et al., 2022; Shibuta et al., 2017).

### 3.3 Point Cloud Representation

Up to this point, our discussion in Section 3 has mainly highlighted graph representations of atomic data, and subsequently, graph neural network architectures. While graphs are a natural way to represent systems of bonded atoms, we also aim to support alternative data representations in our framework, in order to enable development of diverse deep learning architectures. One such example, which is already implemented in Open MatSci ML Toolkit, are atom-centered point cloud representations that are used to achieve translation and rotation equivariance in several recent architectures, including NequIP (Batzner et al.), Geometric Algebra Attention Networks (Spellings, 2021), and Allegro (Musaelian et al., 2022).

Atom-centered point cloud representations often incur a high memory cost, leading us to build our procedure on top of the data abstractions discussed earlier, while only constructing neighborhood point clouds for the most relevant atoms. In the case of OpenCatalyst, this means that we extract the molecular adsorbate (typically a few atoms) and the surface, which—according to prior chemical knowledge—should contribute the most to describing the system as a whole. Furthermore, in each catalyst + molecule system, we randomly sample a number of additional bulk crystal atoms below the surface layer that constitute the substrate for a more holistic description. We construct atomic features using symmetric one-hot embeddings based on the corresponding atomic numbers, which seeks to mirror the treatment for the positions and embed information about the central atom, as well as each non-central atom in the point cloud. Finally, the point cloud data are batched by padding molecule and substrate dimensions with zeros that can subsequently be masked during computation. Given that the point cloud representation shares the same core components as graph datasets, the same pipeline can be used with minimal code changes, largely owing to inheritance patterns (see Figure 5 in Appendix C). The same also applies for different methods of constructing atomic point cloud representations that may be different from atom-centered point clouds we apply in our experiments.

# 4 Experiments & Testing

We designed our experiments to showcase the capabilities of our toolkit highlighted in Section 1. We perform all our experiments on Titan-V GPUs with the number of GPUs and experimental runtimes detailed in each individual experimental section. Concretely, our experiments demonstrate:

- **Seamless Compute Scaling:** We show the benefits of compute scaling across multiple devices in a single node and multi-node setting for different OpenCatalyst tasks described in Section 4.1.

- **Model Enablement:** We implement three new model architectures and show generally reasonable results on OCP-20 tasks described in Section 4.2. Concretely, we provide training results for the following architectures:
    - E(n)-GNN by Satorras et al. (2021) - a rotation-equivariant graph neural network.
    - MegNet by Chen et al. (2019) - a domain specific GNN for materials science applications.
    - Geometric ALgebra Attention Network (GALA) by Spellings (2021) - a short-range equivariant network based on Clifford algebra that applies point cloud representations.

    As shown in Table 1, the success of the new models relies on our compute scaling capabilities and our flexible MLOps data and model pipelines.

All of the models presented have the inductive bias of rotation equivariance; in general, equivariance for functions $f(I)$, $g(I)$ for an entity $I$ is defined as: $f(g(I)) = g(f(I))$. Intuitively this means that the features of an entity transform equally with a given manipulation, such as a rotation. This is particularly useful for property prediction in material compounds, such as the IS2RE and S2EF tasks, where rotation of the entire compound itself does not affect the properties of the compound. Each of the models, however, impart equivariance in different ways through their architecture. We refer the reader to Section 2 and the original papers for further details.

## 4.1 Hardware Scaling Capabilities

| Task | Model Size | Number GPUs | Epoch Time (h) | Experiment Time (h) |
|------|-----------:|------------:|---------------:|--------------------:|
| IS2RE (All) | | | | |
| E(n)-GNN | 72k | 24 | 0.063 | 2.84 |
| MegNet | 986k | 24 | 0.108 | 4.86 |
| Gala | 1.5M | 24 | 1.32 | 59.6 |
| S2EF (2M) | | | | |
| E(n)-GNN | 72k | 32 | 0.404 | 18.2 |
| MegNet | 1.2M | 40 | 1.36 | 61.3 |

Table 1: Compute details for our experiments on IS2RE-All (∼500K Training & ∼25K Validation) and S2EF-2M (2M Training & 1M Validation) indicating the compute requirements to perform effective model training on OCP-20 in reasonable wall-clock time. Both model architecture (Gala > MegNet > E(n)-GNN) and task complexity (S2EF > IS2RE) influence compute needs.

As an illustrative example, we deploy E(n)-GNN using the Open MatSci ML Toolkit to the OCP-20 S2EF task with 200K training samples and 1M validation samples, which is amendable to studying the compute performance one can achieve using our framework. We apply distributed data-parallel training based on the PyTorch library (Li et al., 2020) to scale training across multiple devices on single GPU compute node, as well as multiple GPU compute nodes. As seen in Figure 3, single node scaling to multiple GPUs for the S2EF and IS2RE tasks shows a decreasing benefit as more GPUs are added, asymptotically approaching the apparent limit of parallelizable computation benefits for this particular setting. Epoch training throughput in the multi-node setting for two nodes suggests close to linear scaling. While the overall benefits of compute

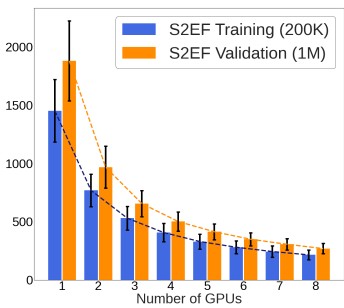 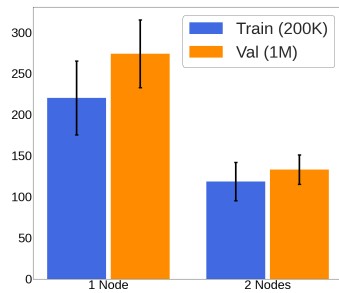 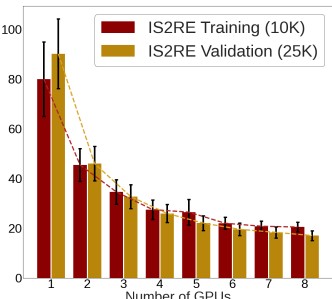

(a) Single Node Scaling on S2EF    (b) Multi-Node Node Scaling on S2EF    (c) Single Node Scaling on IS2RE

Figure 3: Time per Epoch (s) for Multiple Devices on Single Node and 8-Device Multi-Node Setting for the S2EF Task on 200K and 1M Validation Data Split and Single Node Scaling for IS2RE Task on 10K and ∼25K Validation Data Set.

scaling increase with a more intensive task, the overarching compute time, both in core-time and in wall-clock time, also increases making the overall experiment more costly.

## 4.2 OCP-20 Task Performance

We leverage our toolkit to perform training experiments on two large-scale tasks within OCP-20:

1. IS2RE-All: Initial Structure to Relaxed Energy for all data contained in OCP-20 (∼500K training samples). IS2RE is trained on MAE (L1) loss: $\mathcal{L}_\theta^{\text{IS2RE}} = \frac{1}{n}\sum_{i=1}^n |y_i - f_\theta^{\text{energy}}(z_i)|$ with $f_\theta$ being the model parameterized by $\theta$, $y_i$ being the energy labels and $z_i$ being the input data features. The primary aim of this task is to model the absorption energy of the catalyst + adsorbate in the relaxed state, which is generally most relevant for catalytic design applications.

2. S2EF-2M: Structure to Energy & Forces prediction for 2 million training samples. S2EF is also trained on the L1 loss with a separate term for energy prediction and force labels: $\mathcal{L}_\theta^{\text{S2EF}} = \mathcal{L}_\theta^{\text{energy}} + \mathcal{L}_\theta^{\text{force}} = \frac{1}{n}\sum_{i=1}^n |y_i^{\text{energy}} - f_\theta^{\text{energy}}(z_i)| + \frac{1}{n}\sum_{i=1}^n |y_i^{\text{force}} - f_\theta^{\text{force}}(z_i)|$. The forces are obtained by taking the gradient of energy predictions with respect to atomic positions: $f_\theta^{\text{force}} = -\frac{df_\theta^{\text{energy}}(z)}{dx}$ with $x$ being the atomic positions in the system. This requires significantly larger memory and compute compared to the IS2RE task, as well as support for multiple passes of automatic differentiation. The S2EF tasks aims to model the prediction of energy and forces on a single frame thereby training the model to provide a more general approximation of the original chemistry simulation.

We perform the IS2RE-All task for all three models and the S2EF tasks for E(n)-GNN and MegNet. We did not perform S2EF with Gala because the overall memory requirements were too large to fit onto GPU, mainly due to the significant additional memory cost of force predictions. Gala training on IS2RE was already strained in memory utilization given that we applied a batch size of 1 point cloud and performed gradient accumulation across 64 batches to compensate for the small batch size. The full list of relevant hyperparameters can be found in Appendix A.

### 4.2.1 IS2RE-All Task Performance

The IS2RE-All training results in Table 2 and Figures 4a, 4b & 4c show that Gala performs best on training loss and MegNet performs best on in-distribution validation loss. E(n)-GNN performs worst across all in-distribution and out-of-distribution tasks for total task loss. The trend of the training loss, whose deviation corresponds to the loss across various batches in a given epoch, indicates a downwards slope with recurring

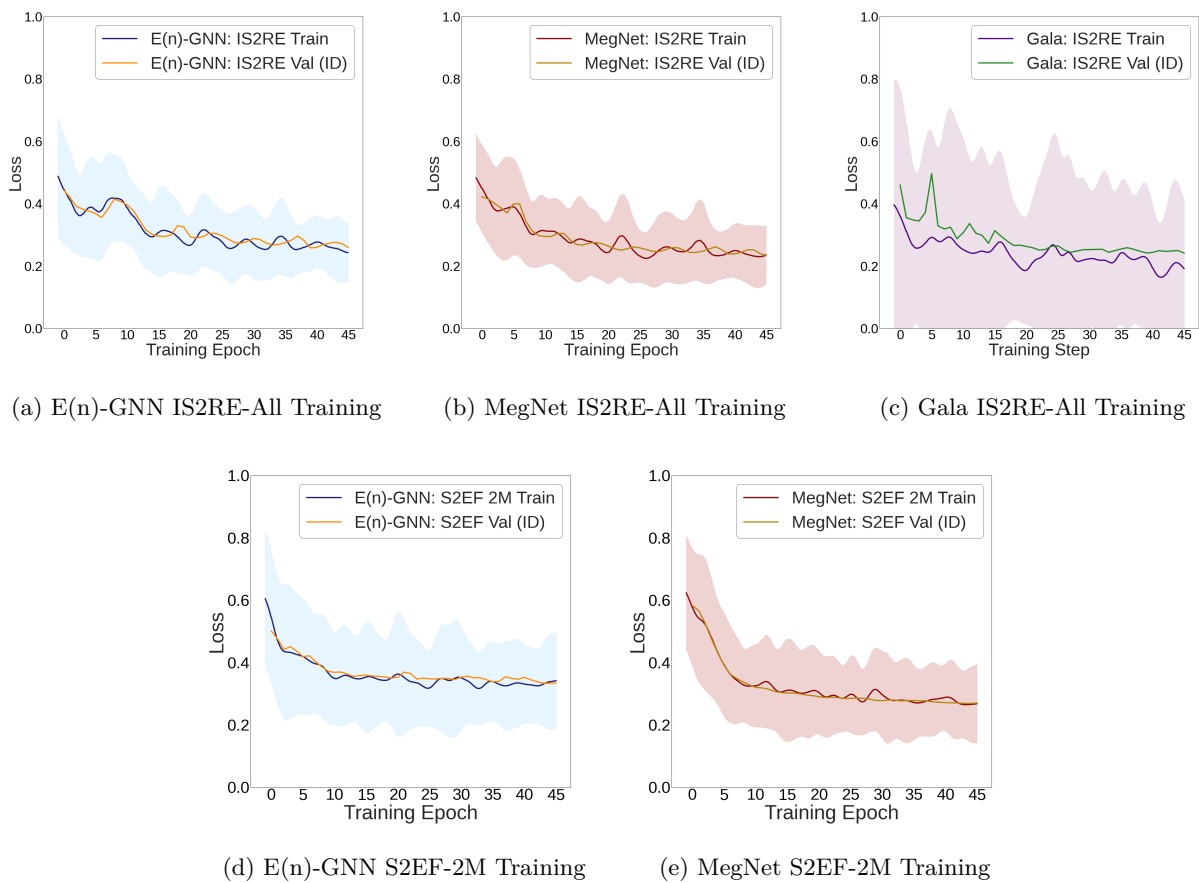

(a) E(n)-GNN IS2RE-All Training  (b) MegNet IS2RE-All Training  (c) Gala IS2RE-All Training

(d) E(n)-GNN S2EF-2M Training  (e) MegNet S2EF-2M Training

Figure 4: Training and in-distribution validation loss curves for the IS2RE-All Task on E(n)-GNN, MegNet and Gala (top row) and the S2EF-2M Task for E(n)-GNN and MegNet (bottom row). In-distribution validation loss generally follows the training loss for all tasks with a somewhat steeper decrease for S2EF than IS2RE. Full results, including out-of-distribution validation losses, are shown in Table 2 and Table 3

| Model | Training | Total Loss ↓ | | | | Energy MAE (eV) ↓ | | | | Energy within Threshold (%) ↑ | | | |
|---|---|---|---|---|---|---|---|---|---|---|---|---|---|
| | Loss | ID | OOD Cat | OOD Ads | OOD Both | ID | OOD Cat | OOD Ads | OOD Both | ID | OOD Cat | OOD Ads | OOD Both |
| **IS2RE Validation Dataset** | | | | | | | | | | | | | |
| **E(n)-GNN** | 0.238 | 0.256 | 0.239 | 0.321 | 0.271 | 0.710 | 0.692 | 0.825 | 0.745 | 3.15 | 3.19 | 1.93 | 2.04 |
| **MegNet** | 0.211 | 0.233 | 0.216 | 0.314 | 0.259 | 0.656 | 0.644 | 0.901 | 0.800 | 3.83 | 3.99 | 1.84 | 1.99 |
| **GALA** | 0.153 | 0.240 | 0.217 | 0.276 | 0.235 | 1.598 | 1.565 | 1.469 | 1.216 | 1.02 | 1.11 | 1.09 | 1.19 |
| **IS2RE Test Dataset** | | | | | | | | | | | | | |
| **E(n)-GNN** | - | - | - | - | - | 0.706 | 0.710 | 0.803 | 0.760 | 3.19 | 3.06 | 1.94 | 1.99 |
| **MegNet** | - | - | - | - | - | 0.653 | 0.668 | 1.138 | 1.123 | 3.90 | 3.55 | 1.74 | 1.60 |
| **GALA** | - | - | - | - | - | 1.585 | 1.565 | 1.649 | 1.475 | 0.97 | 1.05 | 0.80 | 0.86 |

Table 2: Task results for IS2RE (All) with MAE training loss for in-distribution (ID), out-of-distribution catalysts (OOD-Cat), out-of-distribution adsorbates (OOD-Ads), and out-of-distribution catalysts and adsorbates (OOD-Both) for total loss, energy MAE (eV) and energy within threshold (EwT). We highlight the best performing model for each metric. "-" indicates settings that are not applicable for the test data. While GALA mostly has the best performance measured by the loss, MegNet and E(n)-GNN perform better among the energy MAE and EwT.

deviation pattern. This suggests that the networks may find particular data samples more challenging than others as it cycles through the same batches over different epochs. The learning curves for all models also show that the in-distribution validation loss generally follows the training loss, which is expected for this

setting. In the case of out-of-distribution tasks, MegNet performs slightly better than Gala on OOD-Catalyst with Gala performing best on OOD-Adsorbates and OOD-Both. Gala's relatively consistent performance across ID and OOD tasks suggests that Gala may have greater generalization ability compared to E(n)-GNN and MegNet. It is also worth noting that all three models have the most difficulty on the OOD-Adsorbates task, indicating that it is more difficult to generalize for new molecules in the catalyst+molecule system compared to changing the underlying catalyst crystal structure. For the Energy MAE (eV) and EwT metrics MegNet outperforms E(n)-GNN in ID and OOD-Cat, while E(n)-GNN outperforms MegNet in OOD-Ads and OOD-Both. This trends holds across both validation and training data splits.

### 4.2.2  S2EF Task Performance

The total loss values shown in Table 3 and Figures 4d and 4e include errors from both the energy prediction and the force prediction components. Compared to the IS2RE results in Figures 4a, 4b & 4c, the S2EF results in Figures 4d and 4e indicate a stronger downwards slope in both the training and in-distribution validation losses, particularly for MegNet. MegNet outperforms E(n)-GNN across all tasks, including in-distribution and out-of-distribution settings. For non-loss metrics (Energy MAE, EFwT & Force MAE), E(n)-GNN outperforms MegNet for validation data, while MegNet outperforms E(n)-GNN for test data. MegNet's generally stronger performance compared to E(n)-GNN could be related to two factors: 1. Greater representation capacity due to a higher number of model parameters; 2. MegNet's domain specific feature design, including graph-level variables, may be more useful in resolving the greater diversity of data found in the S2EF tasks. The results shown in Table 1 also show the greater compute cost associated with the S2EF-2M task, both in the amount of hardware used as well as the related wall-clock time, driven by larger dataset size and greater task complexity.

| Model | Training | Total Loss ↓ | | | | Energy MAE (eV) ↓ | | | | Energy & Forces within Threshold (%) ↑ | | | | Force MAE (eV/Å) ↓ | | | |
|---|---|---|---|---|---|---|---|---|---|---|---|---|---|---|---|---|---|
| | Loss | ID | OOD Cat | OOD Ads | OOD Both | ID | OOD Cat | OOD Ads | OOD Both | ID | OOD Cat | OOD Ads | OOD Both | ID | OOD Cat | OOD Ads | OOD Both |
| | | | | | | | | **S2EF Validation Dataset** | | | | | | | | | |
| E(n)-GNN | 0.307 | 0.333 | 0.331 | 0.388 | 0.433 | 1.675 | 1.673 | 1.675 | 1.731 | 0.0037 | 0.0031 | 0.0056 | 0.0049 | 0.0811 | 0.0800 | 0.0800 | 0.0943 |
| MegNet | 0.252 | 0.268 | 0.274 | 0.341 | 0.378 | 1.730 | 1.723 | 1.766 | 1.827 | 0 | 0 | 0 | 0 | 0.102 | 0.101 | 0.101 | 0.113 |
| | | | | | | | | **S2EF Test Dataset** | | | | | | | | | |
| E(n)-GNN | - | - | - | - | - | 1.722 | 1.699 | 1.942 | 2.008 | 0 | 0 | 0 | 0 | 0.102 | 0.0994 | 0.103 | 0.118 |
| MegNet | - | - | - | - | - | 1.665 | 1.648 | 1.836 | 1.901 | 0.0038 | 0.0027 | 0.0038 | 0.0016 | 0.0810 | 0.0788 | 0.0802 | 0.0979 |

Table 3: Task results for S2EF (2M) with MAE training loss for in-distribution (ID), out-of-distribution catalysts (OOD-Cat), out-of-distribution adsorbates (OOD-Ads), and out-of-distribution catalysts and adsorbates (OOD-Both) for total loss, energy MAE (eV), forces MAE (eV/Å), and energy within threshold (EwT) as a percentage of the dataset. We highlight the best performing model for each metric. "-" indicates settings that are not applicable for the test data. MegNet outperforms E(n)-GNN for total and for all test data metrics, while E(n)-GNN performs better for validation data on energy MAE, forces MAE and EFwT.

## 5  Discussion

### 5.1  Comparisons with existing models

One of the primary motivations of this paper, and the future work we hope to enable, is the search for new neural network architectures to accelerate materials discovery. We hence contextualize the results in Table 2 and Table 3 by making caveated comparisons with published Open Catalyst results, such as DimeNet++ (Klicpera et al., 2020a) and GemNet-XL (Sriram et al., 2022). In the case of IS2RE, the task measures mean absolute deviation in adsorption energies (in units of eV) calculated with the RPBE density functional (Chanussot* et al., 2021), which estimate the thermodynamic stability of molecules adhering to catalytic surfaces. While direct comparisons are not possible as validation results were not included in earlier papers, the IS2RE MAE results for each model shown in Table 2 are on the order of ∼0.25 eV across multiple validation distributions. The reported test errors for DimeNet++ (0.56 eV, (Chanussot* et al., 2021)) and more recently GemNet-XL (0.38 eV, (Sriram et al., 2022)), suggest that our reported results generally perform worse in Energy MAE.

In the case of S2EF, our comparisons have additional limitations given that our training set of 2M samples is at least an order of magnitude smaller than those applied in other works (Chanussot* et al., 2021; Sriram et al., 2022). Nevertheless, E(n)-GNN and MegNet both present somewhat favorable performance in total loss when compared to GemNet-XL trained on the full S2EF training set (∼133M) by Sriram et al. (2022), who reported OOD-Both joint (sum of energy and forces) test errors of 0.38, compared with 0.43 [E(n)-GNN] and 0.38 (MegNet). We emphasize that, while these are not directly comparable, we believe that our results highlight the potential in discovering novel architectures—particularly parameter efficient ones like E(n)-GNN—and data representations, such as the atom-centered point cloud representation described in Section 3.3, which are facilitated by the Open MatSci ML Toolkit. As the framework matures, we look forward towards scaling up experiments further and enabling direct comparisons with the OCP-20 leaderboard, while also continuing to grow the application of machine learning models for materials discovery applications (Friederich et al., 2021; Chen & Ong, 2022).

## 5.2 Future Work

In this paper, we introduced the Open MatSci ML Toolkit and demonstrated how it can be applied to train advanced geometric deep learning models on different tasks within the OpenCatalyst dataset through seamless compute scaling and automated MLOps. While the primary aim of this paper was to showcase the capabilities of the framework, the challenge of training better and more effective machine learning models on the OpenCatalyst dataset remains. We hope that our results provide a convincing starting point to continue to build on top of the current version of the framework and enable future research in geometric deep learning applied to materials science. Moreover, while the OpenCatalyst remains the largest for materials science, further relevant datasets and benchmarks (Dunn et al., 2020; Jain et al., 2013; Kirklin et al., 2015; Calderon et al., 2015) could be integrated into the Open MatSci ML Toolkit leveraging our flexible and extensible framework to accelerate the development of advanced machine learning tools for materials science.

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

## A Hyperparameters

Example hyperparameters for E(n)-GNN

Table 4: Hyperparameters for E(n)-GNN

| Hyperparameter | Value |
|---|---:|
| MLP hidden dim | 32 |
| MLP output dim | 32 |
| # of EGNN layers | 3 |
| Node MLP dim | $[48, 48]$ |
| Edge MLP dim | $[16, 16]$ |
| Atom position MLP dim | $[64, 64]$ |
| MLP activation | ReLU |
| Graph read out | Sum |
| Node projection block depth | 2 |
| Node projection hidden dim | 128 |
| Node projection activation | ReLU |
| Output block depth | 3 |
| Output hiddem dim | 64 |
| Output activation | ReLU |
| **Optimizer parameters** | |
| Learning Rate | 0.003626 |
| Gamma | 0.6878 |
| Batch Size | 8 |

Example hyperparameters for MegNet

Table 5: Hyperparameters for MegNet

| Hyperparameter | Value |
|---|---:|
| Edge MLP dim | 2 |
| Node MLP dim | 5 |
| Graph variable MLP dim | 9 |
| MLP projection dim | 11 |
| MegNet blocks | 4 |
| MLP hidden dims | $[128, 64]$ |
| MegNet convolution dims | $[128, 128, 64]$ |
| # of S2S layers | 5 |
| # of S2S iterations | 4 |
| Output projection dims | $[64, 16]$ |
| Dropout | 0.1 |
| **Optimizer parameters** | |
| Learning Rate | 0.0001 |
| Gamma | 0.2 |
| Batch Size | 8 |

Example hyperparameters for Gala

Table 6: Hyperparameters for Gala

| Hyperparameter | Value |
|---|---|
| Input dimension | 200 |
| Hidden dimension | 100 |
| Merge function | concat |
| Join function | concat |
| Rotation-invariant mode | full |
| Rotation-covariant mode | full |
| Rotation-invariant value norm | momentum |
| Rotation-equivariant value norm | momentum layer |
| Value function normalization | layer |
| Score function normalization | layer |
| Block-level normalization | layer |
| **Optimizer parameters** | |
| Learning Rate | 0.001 |
| Gamma | 0.8 |
| Batch Size | 1 |

## B  Development Example

A self-contained python script running the full pipeline on one of our dev-sets is shown below:

```python
""" Sample Python Script Without Imports """

# Define Parameters
BATCH_SIZE = 8
NUM_WORKERS = 4
REGRESS_FORCES = False
epochs = 5

# Model configuration for MegNet
model_config = {
    "edge_feat_dim": 2,
    "node_feat_dim": 5,
    "graph_attr_dim ": 9,
    "dim": 1,
    "num_blocks": 4,
    "hiddens": [128, 64]
    "conv_hiddens": [128, 128, 64]
    "s2s_num_layers": 5,
    "s2s_num_iters": 4,
    "output_hiddens": [64, 16],
    "is_classification": False,
    "dropout": 0.1,
}

# use default settings for MegNet
megnet = MegNet(**model_config)

# use the GNN in the LitModule for all the logging, loss computation, etc.
model = S2EFLitModule(megnet, regress_forces=REGRESS_FORCES, lr=1e-3, gamma=0.1)
data_module = S2EFDGLDataModule.from_devset(
    batch_size=BATCH_SIZE, num_workers=NUM_WORKERS
)

```

```
36  # alternatively, if you don't want to run with validation, just do S2EFDGLDataModule.
        from_devset
37  data_module = S2EFDGLDataModule(
38      train_path=s2ef_devset,
39      val_path=s2ef_devset,
40      batch_size=BATCH_SIZE,
41      num_workers=NUM_WORKERS,
42  )
43
44  trainer = pl.Trainer(accelerator="gpu", strategy="ddp", devices=2, max_epochs=epochs)
45  trainer.fit(model, datamodule=data_module)
```

Listing 1: Self-Contained Example Script With Scalable Devices

As can be seen in the definition of `trainer`, this short script already performs training on two GPUs with users being able to change the `devices` variable to adjust the numbers of GPUs they want to leverage for distributed training on a single node.

## C   Data pipeline abstraction

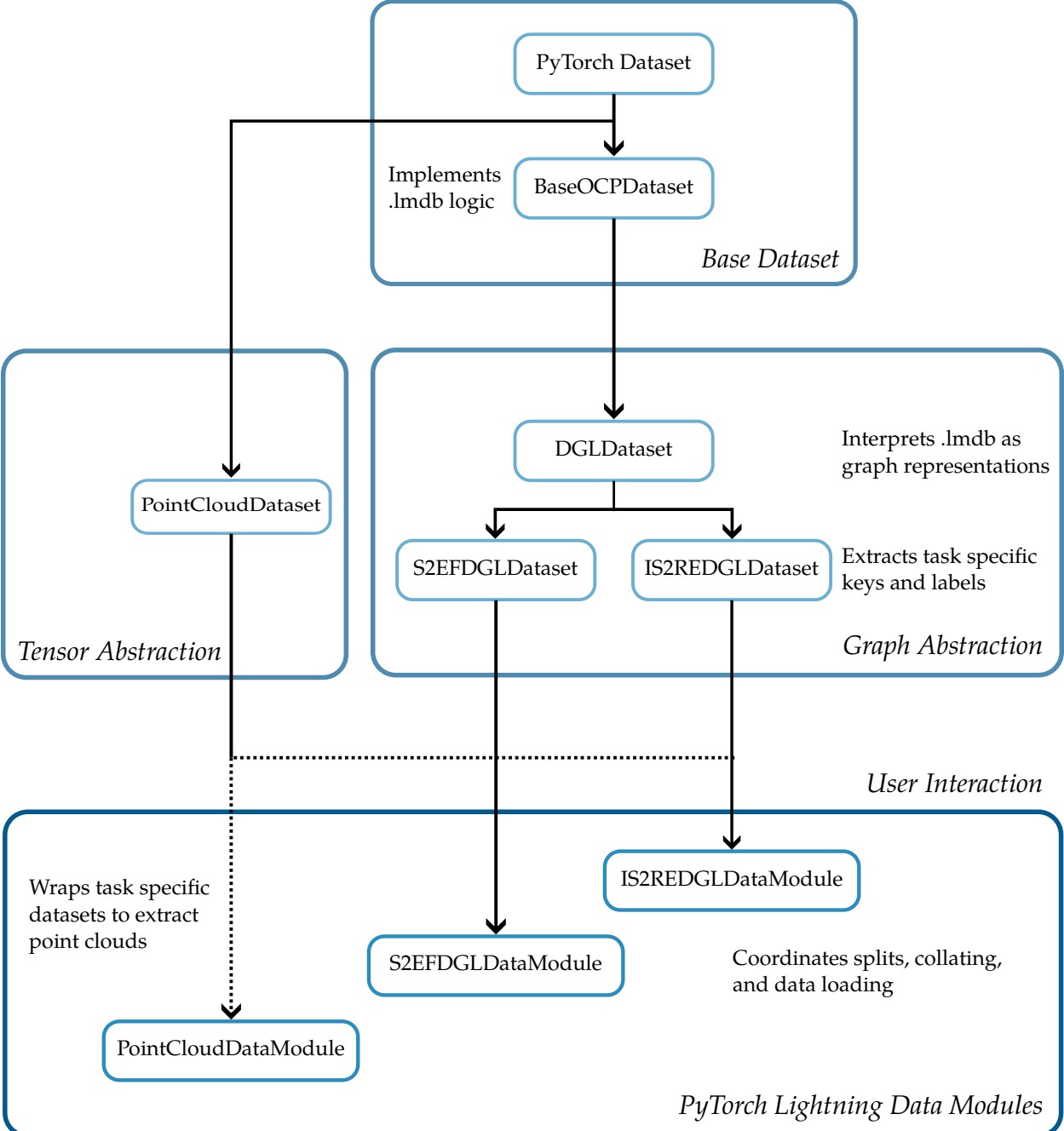

Figure 5: Inheritance diagram for the data abstraction in Open MatSci ML Toolkit. The main user interaction layer is presented at the bottom, corresponding to subclasses of `LightningDataModule`. Arrows denote directional relationship between the classes; the dashed line indicates that the `PointCloudDataset` wraps the task specific datasets, whereby the user is provided with a sampled point cloud representation of the original graphs.

