# OpenReview forum: "The Open MatSci ML Toolkit: A Flexible Framework for Machine Learning in Materials Science"
_TMLR — Accepted by TMLR_

### Review · Reviewer_3e5y · 2023-04-28

**Summary Of Contributions:**

The authors present The Open MatSci ML Toolkit, which is a framework for developing ML models for materials, and in particular the Open Catalyst dataset. They give a detailed discussion of the considerations behind their design decisions, and how they make both starting and scale-up easy. The paper finishes with experiments comparing existing architectures to showcase what the framework can do.

**Audience:**

Yes

**Broader Impact Concerns:**

I see no unaddressed ethical concerns.

**Claims And Evidence:**

Yes

**Requested Changes:**

As a start, I'd request the authors to fix the various small issues mentioned above, in particular the repo link.

**Strengths And Weaknesses:**

=== Strengths ===

The framework seems reasonably designed, and the authors give a good amount of attention to detailed considerations, like differences between relying on DGL vs PyG. While I don't work with Open Catalyst personally, the toolkit seems like a useful thing to have, especially for somebody just starting out. The paper is also very well-written and clear.



=== Weaknesses ===

The authors spend a lot of time outlining the MLOps improvements (like seamless transition from CPU to GPU), but most of these benefits just come from using `pytorch_lightning`. This is not a weakness in itself (using existing frameworks is good, we should not try to reinvent the wheel!), but examining things more closely I'm not sure if the "volume" of work done here is enough for a paper at TMLR.



=== Other comments & questions ===

- (O1): Repo link does not work for me, as it claims the repository has expired, thus I couldn't take the code into account during my review.

- (O2): In pseudocode on page 7, I was confused why `data` is assigned to twice. After reading the surrounding text I realized the pseudocode is simply showing two ways of creating a `Data` object, and is not supposed to be interpreted as a single chunk of code. Maybe modify the comments in that snipped to include something like "Option 1:" and "Option 2:" for clarity.



=== Nitpicks ===

- In the caption of Figure 2, the font style of the last letter of "LightningDataModules" is not consistent with how this is handled in the other such cases.

- "grow the application of machine learning models for materials discovery applications" - a bit repetitive.

---

> ### Author Response · Authors · 2023-05-09
> **Response to Reviewer 3e5y**
>
> Thank you for your feedback. Here are our responses to your questions:
> 1.	As stated in our general response, we contribute:
> * **Designing and building a flexible software framework for the OpenCatalyst dataset:** While PyTorch-Lightning serves as an engine that enables us to use best-in-class MLOps, we still need to build the framework around it to be able to leverage it effectively for our desired ML tasks. As described in Section 3, this includes abstraction for the data (defining data structure and format), the model (defining a re-usable model API to ensure new models can be added easily) and training loops (defining all the optimizations to be used, including double backwards passes for force computation in the S2EF task, and relevant logging capabilities that are written on top of PyTorch-Lightning as Callbacks).
> * **Redefining and building relevant datasets and abstracts:** As described in Section 3.2 and Section 3.3, we introduce two novel data structures for the OpenCatalyst dataset (DGL graphs and Point Clouds). Both of these new data definitions necessitate that we redefine and transform the original OpenCatalyst data into new format that are compatible with the models and frameworks in the Open MatSci ML Toolkit. As such, we created data conversion routines to transfer the original data to our more flexible format and have our own dataset release for users to download directly.
> * **Enabling new machine learning model architectures:** As described in Section 3.3, our framework easily enables a new class of geometric deep learning based on our point cloud data structure. This will make it easier for researchers to develop new types of geometric deep learning models that can work across different representations all within the Open MatSci ML Toolkit.
> 2.	Thank you for pointing this out. We fixed and updated the repo link and it should work now.
> 3.	Thank you. We will add that clarification to the code snippet.
> 4.	Nitpick – thank you for pointing these out. We will update these in the next paper version.

---

### Review · Reviewer_6eMk · 2023-05-02

**Summary Of Contributions:**

The authors have presented an open-source software toolkit named Open MatSci ML, which builds on top of the OpenCatalyst project (OCP20) and provides scalable deep learning solutions for various catalyst modeling tasks. The OpenCatalyst Dataset is currently the largest open-source dataset in the field of computational materials sciences. The original OCP20 paper demonstrated the use of state-of-the-art graph neural network models to solve three tasks indicative of day-to-day catalyst modeling. In this work, the authors refactored OCP20 in PyTorch Lightning and Deep Graph Library (DGL) to provide ease of use for researchers, scalable computation across devices, faster training, and better support for molecular graphs. They also explored a few new model architectures, including E(n)-GNN, MegNet, and Geometric Algebra Attention Network (Gala), which showed competitive results on OCP20 tasks. As an extension of the OpenCatalyst project, the graph deep learning framework presented in this work has the potential to accelerate materials discovery and enable catalyst modeling for clean energy applications.

**Audience:**

Yes

**Claims And Evidence:**

Yes

**Requested Changes:**

1. The authors could expand on the rationale behind choosing DGL and point cloud representations. Additionally, they could provide further discussions on how these representations enable new model architecture and lead to gains in prediction accuracy.
2. The discussion in the task performance section could be expanded further. The authors should discuss the reasons that make these models suitable for deep graph neural networks and catalyst modeling.
3. In Section 4.1, the author should mention the specifications of the GPU nodes used for the experiments
4. Table 2 displays the in-distribution validation loss and out-of-distribution validation loss. I assume the latter is the independent test set. How does the test set performance compare to the original three models presented in the OCP20 paper? The original test set metrics should also be included in the table and compared to the metrics obtained in this paper. Additionally, each task in the OCP20 has multiple metrics representing the key structural and electronic properties of the catalysts. Does the new model predict other metrics as well? For instance, for the IS2RE task, how do the models perform on the EwT metric, going beyond the energy MAE loss?
5. In the future work section, the author should discuss how easy it would be to extend the framework to other material science or catalyst datasets. What is the required data format?  The author could provide guidelines for expanding the OCP20 dataset. After all, the OCP20 dataset only consists of molecular adsorptions onto surfaces. Alloys, reactions, transition states, and so on are also of interest to material scientists.


By addressing these additional points, the authors can further strengthen their papers and make this work more appealing to its target audience, i.e., researchers in the AI, chemistry, and materials science community.


**Strengths And Weaknesses:**

The strengths of this work include:

-	The paper is well-written and the language is easy to understand

-	The introduction provides brief summary of why catalysts and materials discovery using AI/ML is important. This makes this work appealing to readers outside the field of chemistry and materials science.

-	It utilizes faster and more scalable deep learning frameworks such as PyTorch Lightning and DGL to enable cross-platform optimization and sampling-based training. This effort essentially lowers the barrier of entry to use the OCP20 dataset. The code could potentially run on a single-node GPU or data center CPUs with built-in data parallelism.

-	New model architectures are explored for OCP20 tasks, and they show promising results. The experiments ran on the new framework were able to finish within a reasonable amount of time (shorter than 60 hours).

The weaknesses of this work include:

-	Some context and details of OCP20 tasks are missing or not well-elaborated in this paper. For example, the three tasks with the abbreviations of OOD Cat, OOD Ads, and OOD both were not explained in this paper. This omission makes it difficult to read this paper as a standalone study without referring to the original OCP20 paper.
-  There is a lack of connection between the choice of methods and the gains in experimental results. For instance, it is unclear how the refactor in DGL and the point of cloud representation would result in any performance gains in modeling molecular graphs. Additionally, the authors should expand on the rationale behind the new model architecture, namely, why those models were chosen and why they are suitable for the OCP20.
-	In the discussion part of task performance, the authors only briefly mention the models that perform well on certain tasks. However, they do not delve into the potential causes of these performance differences. For researchers in the catalysis field, understanding the reasons why some models perform well can be even more significant than achieving high scores in benchmarking tests. This knowledge can provide valuable insights into the underlying mechanisms of catalysis and lead to the development of more effective catalysts. Furthermore, understanding the reasons behind model performance can aid in improving the accuracy and reliability of predictions, leading to more efficient modeling approaches.

---

> ### Author Response · Authors · 2023-05-09
> **Response to Reviewer 6eMk**
>
> Thank you for your feedback. Here are our responses to your questions:
>
> 1.	Thank you for pointing this out. We will add a deeper discussion in the paper on this. Essentially, the main reason is to enable researcher to create new modeling techniques based on both graph and point cloud representations. The fact that we can train a new class of models (like GALA) already show the promise of enabling data structures. These new data structures can be scientifically meaningful: as described in prior work in the literature [1], models based on near-neighbor point clouds—in contrast to message-passing with GNNs—can have a smaller domain and induce shorter-range interactions that we may expect to be easier to learn for a given training data size or model capacity.
>
> 2.	Thank you for pointing this out. We will add a deeper discussion in the paper about this. We choose our models to compare across: 1. A rotation-equivariant GNN; 2. A domain-specific GNN; 3. A novel, rotation-equivariant architecture based on point cloud representations.
>
> 3.	Yes – we will add these details into the next draft. Most of the GPUs used were Titan-V.
>
> 4.	Thank you for this suggestion. We agree this is relevant information that will the paper stronger and will modify our results to include the original test metrics from [2], the original OCP paper. We will also calculate relevant additional metrics, such as EwT, to get a clearer picture of model performance.
>
> 5.	We designed the framework to make it easy to add new datasets as well as new models. These were one of the primary motivations for constructing the toolkit as we did since we do not want it to be limited only to the OCP dataset.
>
>
> [1] Musaelian, A., Batzner, S., Johansson, A., Sun, L., Owen, C. J., Kornbluth, M., and Kozinsky, B. Learning local equivariant representations for large-scale atomistic dynamics. arXiv preprint arXiv:2204.05249, 2022.
>
> [2] Chanussot*, L., Das*, A., Goyal*, S., Lavril*, T., Shuaibi*, M., Riviere, M., Tran, K., Heras-Domingo, J., Ho, C., Hu, W., Palizhati, A., Sriram, A., Wood, B., Yoon, J., Parikh, D., Zitnick, C. L., and Ulissi, Z. Open
> catalyst 2020 (oc20) dataset and community challenges. ACS Catalysis, 2021. doi: 10.1021/acscatal.0c04525.

---

### Review · Reviewer_347i · 2023-05-06

**Summary Of Contributions:**

The paper introduce the Open MatSci ML Toolkit: framework to apply deep learning models and methods on tasks from the field of materials science, and specifically from OpenCatalyst Dataset.

**Audience:**

Yes

**Broader Impact Concerns:**

No.

**Claims And Evidence:**

Yes

**Requested Changes:**

Follow my answers in the previous section.

**Strengths And Weaknesses:**

Strengths

1. The framework enables to use both graph representation, point cloud representation and other (futuristic) representation of the data.

2. The selection of the base frameworks (DGL and PyTorch Lighting) is well justified.


Weaknesses

1. From my understanding it seems that most of the work of model and data abstractions, scaling, logging and more is done by DGL and PyTorch Lightning. Please clearly state your exact contribution to the pipeline.

2.  The experiments lack comparison to previous works. Please add timing comparisons between your framework to previous works.

---

> ### Author Response · Authors · 2023-05-09
> **Response to Reviewer 347i**
>
> Thank you for your feedback. Here are our responses to your questions:
>
> 1.	As stated in our general response, we contribute:
> * **Designing and building a flexible software framework for the OpenCatalyst dataset:** While PyTorch-Lightning serves as an engine that enables us to use best-in-class MLOps, we still need to build the framework around it to be able to leverage it effectively for our desired ML tasks. As described in Section 3, this includes abstraction for the data (defining data structure and format), the model (defining a re-usable model API to ensure new models can be added easily) and training loops (defining all the optimizations to be used, including double backwards passes for force computation in the S2EF task, and relevant logging capabilities that are written on top of PyTorch-Lightning as Callbacks).
> * **Redefining and building relevant datasets and abstracts:** As described in Section 3.2 and Section 3.3, we introduce two novel data structures for the OpenCatalyst dataset (DGL graphs and Point Clouds). Both of these new data definitions necessitate that we redefine and transform the original OpenCatalyst data into new format that are compatible with the models and frameworks in the Open MatSci ML Toolkit. As such, we created data conversion routines to transfer the original data to our more flexible format and have our own dataset release for users to download directly.
> * **Enabling new machine learning model architectures:** As described in Section 3.3, our framework easily enables a new class of geometric deep learning based on our point cloud data structure. This will make it easier for researchers to develop new types of geometric deep learning models that can work across different representations all within the Open MatSci ML Toolkit.
> 2.	Could you provide more detail on what you are looking for? In Figure 3 (Section 4), we show timing based on our own framework. Running the same models with the original repo would require porting them to PyG, which would introduce additional confounding factors.

---

### Author Response · Authors · 2023-05-09
**General Reviewer Response**

We thank the reviewers for their feedback and suggestions. We are encouraged by the fact that all reviewers found our framework to be a well-made and a valuable addition to the research community working at the intersection of AI and materials & chemistry. We are also encouraged by the fact that the reviewers view the addition of PyTorch-Lightning and DGL, along with our point cloud representations, to be an enabling addition to the machine learning community and that our easy-to-use toolkit makes it easy for researchers to get started with the Open Catalyst dataset.

Reviewer 3e5y and Reviewer 347i asked us to clarify our contributions given the reliance of our framework on PyTorch-Lightning and DGL. As described in Section 3.1, we intentionally offload most of the MLOps to PyTorch-Lightning to obtain the benefits of robustness, uniformity and scalability. As also mentioned by Reviewer 3e5y, this allows to not have to “reinvent the wheel” and focus on our primary contributions, which are:

* **Designing and building a flexible software framework for the OpenCatalyst dataset:** While PyTorch-Lightning serves as an engine that enables us to use best-in-class MLOps, we still need to build the framework around it to be able to leverage it effectively for our desired ML tasks. As described in Section 3, this includes abstraction for the data (defining data structure and format), the model (defining a re-usable model API to ensure new models can be added easily) and training loops (defining all the optimizations to be used, including double backwards passes for force computation in the S2EF task, and relevant logging capabilities that are written on top of PyTorch-Lightning as Callbacks).
* **Redefining and building relevant datasets and abstracts:** As described in Section 3.2 and Section 3.3, we introduce two novel data structures for the OpenCatalyst dataset (DGL graphs and Point Clouds). Both of these new data definitions necessitate that we redefine and transform the original OpenCatalyst data into new format that are compatible with the models and frameworks in the Open MatSci ML Toolkit. As such, we created data conversion routines to transfer the original data to our more flexible format and have our own dataset release for users to download directly.
* **Enabling new machine learning model architectures:** As described in Section 3.3, our framework easily enables a new class of geometric deep learning based on our point cloud data structure. This will make it easier for researchers to develop new types of geometric deep learning models that can work across different representations all within the Open MatSci ML Toolkit.


We will address further reviewer comments in more details in individual responses.

---

### Decision · Action_Editors · 2023-07-06

**Recommendation:** Accept with minor revision

**Comment:**

All reviewers agreed that the companion paper to the library is a useful contribution to the field. All reviewers were also satisfied with the answers provided by the authors. Authors should incorporate the additional performance metrics of the OpenCatalyst Dataset paper as discussed with reviewer 6eMk. The authors should also provide a link to the repo and could include the hardware used for the experiments.

**Audience:**

This paper is of interest to researches that leverage machine learning in (computational) material sciences. While this could be viewed as a small group of TMLR's audience, it is still relevant to the community and could raise aware to this application domain.

**Claims And Evidence:**

This work details a Python-based open-source library that facilitates the use of deep learning in the field of computational material sciences. The toolkit integrates the OpenCatalyst Dataset, the largest collection of data sets in this domain. The library also leverages PyTorch Lightning and Deep Graph Library to further accelerate research in the area by offering a broad set of scalable tools and methods, as well as a scalable execution backend.